Gastric intestinal metaplasia subtypes and the effects of c-Myc expression on severity

Yang Qinglu 1
Lian Lingzhi 2
Shen Jingying 2
Cao Qin 1
Wang Xuewei 18801809886@163.com 1
Hu Pingping pt10848@shutcm.edu.cn 2
1 Department of Gastroenterology, Putuo Hospital, Shanghai University of Traditional Chinese Medicine , Shanghai , China
2 Department of Pathology, Putuo Hospital, Shanghai University of Traditional Chinese Medicine , Shanghai , China
Anson Lesley
Electronic publication date: 2025 Oct 31
Publication date: 2025
Volume: 13
Electronic Location ID: e20257
Received 2025 May 5; Accepted 2025 Sep 26
Copyright: ©2025 Yang et al.
Copyright year: 2025
Copyright holder: Yang et al.
License: This is an open access article distributed under the terms of the Creative Commons Attribution License, which permits unrestricted use, distribution, reproduction and adaptation in any medium and for any purpose provided that it is properly attributed. For attribution, the original author(s), title, publication source (PeerJ) and either DOI or URL of the article must be cited.
License URL: https://creativecommons.org/licenses/by/4.0/

Keywords: Gastric intestinal metaplasia, Gastric intestinal metaplasia subtype, C-Myc, Gastric cancer

Funding: The Shanghai Putuo District clinical characteristic disease construction project 2023tszb01 This work was supported by the Shanghai Putuo District clinical characteristic disease construction project (2023tszb01). The funders had no role in study design, data collection and analysis, decision to publish, or preparation of the manuscript.

==============================
Background

The association between gastric intestinal metaplasia severity grades, histological subtypes, and oncogenic potential remains unclear. This study explored gastric intestinal metaplasia (GIM) subtypes and c-Myc protein expression across mild, moderate, and severe GIM cases.

Methods

A total of 180 paraffin-embedded gastroscopy biopsy samples from patients diagnosed with atrophic gastritis were selected, with 60 cases each of mild, moderate, and severe GIM. Alcian blue-Periodic acid-Schiff (AB-PAS) and high iron diamine (HID) staining were used to classify GIM into types I-III. Immunohistochemistry was performed to assess c-Myc expression, with low, moderate, and high expression defined as the percentage of c-Myc-positive cells in the GIM area of <15%, 15–40%, and ≥ 40%, respectively. Spearman and Kruskal–Wallis tests were used to analyze the correlation between GIM severity, GIM subtype, and c-Myc expression.

Results

GIM was predominantly diagnosed in middle-aged and elderly individuals. Regarding the subtype, 53.89% were type II, 25.56% were type III, and 20.56% were type I. Low c-Myc expression was present in 47.78% of cases, moderate expression in 36.67%, and high expression in 15.56%. Neither the severity of GIM nor its subtype or c-Myc expression level was correlated with age or sex. Type III GIM accounted for approximately 10% of mild-to-moderate cases, whereas > 50% of severe GIM cases were type III. A positive correlation was found between GIM severity and subtype (rs = 0.376, P < 0.05). There was no significant correlation in c-Myc expression across different GIM severities. From type I to type III GIM, the proportion of low c-Myc expression increased and that of high expression decreased, whereas that of moderate expression remained almost unchanged. A negative correlation was observed between the GIM subtype and c-Myc expression (rs = −0.148, P < 0.05).

Conclusion

GIM incidence increases with age; however, the histological severity of GIM (as defined by the extent of mucosal gland involvement) within a single biopsy sample does not show a corresponding increase with age. The more severe the GIM is, the greater the proportion of type III GIM cases present. c-Myc expression did not correlate with GIM severity. Conversely, as the GIM subtype becomes more advanced (from type I to type III), c-Myc expression decreases.

Introduction

In 2022, gastric cancer ranked fifth among the world’s leading causes of cancer-related death and fifth in terms of cancer incidence rates (Bray et al., 2024). In 2022, the incidence of gastric cancer in China was 35.87 per 100,000, ranking under lung, colorectal, thyroid, and liver cancer, with a mortality rate of 26.04 per 100,000, ranking under lung and liver cancer. Histologically, gastric cancer can be divided into intestinal and diffuse types (Zheng et al., 2024). Correa was the first to propose a cascade for the development of intestinal-type gastric cancer: superficial gastritis, atrophic gastritis, gastric intestinal metaplasia (GIM), dysplasia, and gastric cancer (Correa, 1988). Gastric mucosal atrophy and intestinal metaplasia (IM) are considered the precancerous conditions of intestinal-type gastric cancer, and Helicobacter pylori infection is the most important causative factor (Sugano, Moss & Kuipers, 2023). At present, according to the new Sydney system guidance, the pathological diagnosis of chronic gastritis defines the severity of GIM as—(none), + (mild), ++ (moderate), or +++ (severe) (Capelle et al., 2010a; Dixon et al., 1996; Rugge & Genta, 2005). The greater the severity of GIM in patients with atrophic gastritis is, the greater the risk of gastric cancer development, which is the main cause of anxiety in these patients. However, the new Sydney system does not indicate to what extent the severity of GIM reflects the risk of cancer development. GIM can be further classified into three types by histology and mucus staining: type I (complete GIM), type II (incomplete small intestinal metaplasia), and type III (incomplete colonic metaplasia) (Jass & Filipe, 1981). The risk of gastric cancer is thought to be significantly greater in type III GIM than in type II GIM, which in turn is significantly greater than that in type I GIM (Shah et al., 2020). However, in clinical practice, only a small percentage of type III GIMs evolve into gastric cancer. In recent years, single-cell transcriptome and whole-exome analyses have suggested that c-Myc plays an important role in the transition from GIM to intestinal-type gastric cancer (Huang et al., 2023; Kumagai et al., 2022). Therefore, this study investigated GIM subtypes and c-Myc expression across different GIM severities by histochemical staining and immunohistochemistry and explored the correlations among the three factors to provide new suggestions and ideas for clinical decision-making and pathogenesis research on GIM.

Materials & Methods

The study was granted by the Ethics Committee of Putuo District Center Hospital, Shanghai (Putuo Hospital,Shanghai University of Traditional Chinese Medicine) (Ethics Approval No.: PTEC-A-2025-8(S)-1).

General information

From October 1, 2024, to January 31, 2025, we conducted a screening of paraffin-embedded endoscopic gastric biopsy specimens archived in the Department of Pathology at Putuo Hospital affiliated with Shanghai University of Traditional Chinese Medicine, spanning the years 2022 to 2023. Based on the predefined study criteria, a total of 320 eligible specimens were identified. Among these, 180 patients provided signed informed consent forms. Subsequently, relevant histopathological staining analyses for these 180 specimens were initiated on March 15, 2025. The samples were divided into three groups (60 per group) based on GIM severity according to the New Sydney System (Dixon et al., 1996): mild (+): GIM involves <1/3 of the mucosal glandular layer; moderate (++): GIM involves 1/3 − 2/3 of the mucosal glandular layer; severe (+++): GIM involves>2/3 of the mucosal glandular layer. Each GIM case was selected after an independent review by two pathologists, and the diagnosis was consistent. As this is a retrospective study utilizing archived biopsy specimens, systematic H. pylori testing (current or past infection status) was not uniformly performed or documented in the medical records for all included patients prior to biopsy archiving. Consequently, data on H. pylori status were incomplete and unavailable for reliable analysis in this cohort. The primary focus of this study was the relationship between GIM severity, subtype, and c-Myc expression within the metaplastic lesions themselves. While recognizing H. pylori’s established role in gastritis and GIM pathogenesis, its influence was not the central investigative aim here. Future prospective studies will incorporate systematic H. pylori testing to address its potential interactions. The patients were 34–79 years of age (mean 63.8 ± 8.9 years); 84 were male, and 96 were female.

Reagents

The mouse anti-human c-Myc primary antibody and immunohistochemical secondary antibody kit were obtained from Beijing Zhongshan Jinqiao Biotechnology Co., Ltd. Alcian blue-periodic acid-Schiff (AB-PAS) and high-iron diamine (HID) staining solutions were obtained from Shanghai Suhai Medical Equipment Co., Ltd.

AB-PAS staining

The paraffin-embedded tissue was cut into 4–5 µm sections, routinely deparaffinized in water, immersed in 1% AB staining solution (prepared with 3% glacial acetic acid, pH 2.5), stained at room temperature for 30–60 min, and then washed with water. Then, the sections were immersed in 0.5–1% periodic acid solution for oxidation for 5–10 min, rinsed with distilled water, immersed in Schiff reagent, and stained in the dark for 15–30 min. The sections were subsequently rinsed with sulfurous acid aqueous solution, dehydrated with different concentrations of ethanol, cleared in xylene, and mounted with neutral gum.

AB-HID staining

As with the AB-PAS staining step, the paraffin-embedded sections were routinely deparaffinized in water, immersed in freshly prepared HID staining solution, stained at room temperature for 12–24 h, and then washed with water. The sections were immersed in 1% AB staining solution (prepared with 3% glacial acetic acid, pH 2.5), stained at room temperature for 30–60 min, and then washed with water. Finally, the slices were sequentially dehydrated with gradient ethanol, cleared in xylene, and mounted with neutral gum.

c-Myc immunohistochemistry

The multimer-labeled immunohistochemical staining method was performed as follows: paraffin-embedded sections were deparaffinized with xylene and then hydrated in gradient ethanol sequentially. High-temperature antigen retrieval was conducted in citrate buffer (pH 6.0), and the samples were then washed in PBS three times for 5 min each. The sections were blocked in 10% goat serum at room temperature for 30 min, and then primary antibody was added and incubated overnight at 4 °C. After the samples were washed in PBS, multimeric HRP-labeled goat anti-mouse secondary antibody (EnVwasion complex) was added, and the samples were incubated at room temperature for 30 min. The cells were then washed in PBS. The sections were subsequently subjected to DAB staining, hematoxylin counterstaining, gradient ethanol dehydration, xylene clearing, and neutral gum mounting.

GIM subtype and criteria for c-Myc expression

GIM can be classified into three types using mucus staining. Type I (complete intestinal metaplasia): AB+, PAS-, HID-; morphologically similar to the small intestine or colon mucosal epithelium, with goblet, absorptive, and Paneth cells. Goblet cells contain sialic acid or sulfate mucus, which can be stained blue by AB, and absorbent cells do not contain mucus. Type II (incomplete small intestinal metaplasia) AB+, PAS+, and HID-; in addition to goblet cells, columnar cells secreting neutral mucus can be seen, which can be stained red by PAS. Paneth cells are rarely seen. Type III (incomplete colonic metaplasia): AB+, PAS-, and HID+; in addition to goblet cells, columnar cells secreting sulfate mucus can be seen, stained brownish black by HID. Paneth cells are rarely seen. Histologically, the glandular epithelial cells are slightly irregular, and the GIM glands are slightly tortuous and branched.

Criteria of c-Myc expression: The percentage of c-Myc-positive cells in the GIM area in each slice was evaluated according to the general practice of immunohistochemistry (Thunnissen et al., 2017). In our preliminary experiment, the percentage of c-Myc-positive cells in 80% of mild superficial gastritis patients (with mild chronic inflammation and negative results for all other pathological indicators) was <15%, and combined with the expression frequency in the current study group, we defined <15% as low expression, 15–40% as moderate expression, and ≥40% as high expression.

Statistical analysis

SPSS 26.0 software (IBM Corp., Armonk, NY, USA) was used for the statistical analyses, and the data are expressed as numbers and percentages. The nonparametric Spearman test was used for bidirectional ordinal variables, and the nonparametric Kruskal–Wallis test was used for unidirectional ordinal variables. P < 0.05 was considered to indicate significance.

Results

General information

Among the 180 enrolled GIM cases, there were 60 cases each of mild (+), moderate (++), and severe (+++) GIM. The cohort comprised slightly more females (53.33%) than males (46.67%), primarily middle-aged and elderly individuals. Type II GIM was the most prevalent subtype (53.89%), followed by Type III (25.56%), while Type I was the least common (20.56%). Regarding c-Myc expression, low expression (<15% positive cells) was observed in the majority of cases (47.78%), followed by moderate expression (15–40%, 36.67%), with high expression (≥40%) being the least frequent (15.56%) (Table 1).

Table 1 General information of 180 GIM cases.

Demographic and pathological characteristics of the study cohort (n = 180). GIM severity was classified according to the New Sydney System based on the extent of glandular layer involvement in the biopsy sample. GIM subtypes were determined by combined AB-PAS and HID histochemical staining. c-Myc expression levels were categorized based on the percentage of positive cells within the GIM area.

Category	Number of cases (%)	
Sex		
Male	84 (46.67)	
Female	96 (53.33)	
Age		
<65 years	81 (45.00)	
≥65 years	99 (55.00)	
Severity of GIM		
Mild (+)	60 (33.33)	
Moderate (++)	60 (33.33)	
Severe (+++)	60 (33.33)	
Intestinal typing		
Type I (complete intestinal metaplasia)	37 (20.56)	
Type II (incomplete small intestinal metaplasia)	97 (53.89)	
Type III (incomplete colonic metaplasia)	46 (25.56)	
Expression of c-Myc (percentage of positive cells)		
Low expression (<15%)	86 (47.78)	
Moderate expression (15–40%)	66 (36.67)	
High expression (≥40%)	28 (15.56)	
Notes.

GIM gastric intestinal metaplasia

Staining manifestations

The manifestations of HE staining for different GIM severities, mucus staining for the GIM subtype, and c-Myc immunohistochemical staining are shown in Fig. 1.

Figure 1 Representative photomicrographs of pathological staining in gastric mucosa biopsies (×100 magnification).

(A–C) Hematoxylin-Eosin (HE) staining demonstrating GIM severity. (A) Mild GIM (+): Involves ≤ 1/3 of the mucosal glandular layer. (B) Moderate GIM (++): Involves 1/3-2/3 of the mucosal glandular layer. (C) Severe GIM (+++): Involves>2/3 of the mucosal glandular layer. (D–F) Combined histochemical mucus staining for GIM subtyping. (D) Type I GIM: Goblet cells stain blue with Alcian Blue (AB+), absorptive cells are Periodic acid-Schiff negative (PAS-), and high iron diamine negative (HID-). (E) Type II GIM: Goblet cells AB+, columnar cells stain red with PAS (PAS+), HID-. (F) Type III GIM: Goblet cells AB+, columnar cells stain brownish-black with HID (HID+), some goblet cells may also be HID+. PAS staining is typically negative or weak in goblet cells (PAS-). (G–I) Immunohistochemical staining for c-Myc protein within the GIM area. (G) Low expression (≤ 15% positive cells). (H) Moderate expression (15–40% positive cells). (I) High expression (≥ 40% positive cells).

Correlation between GIM severity and age/sex

The number of GIM cases was higher in the ≥65 years age group (99 cases) compared to the <65 years group (81 cases). However, the distribution of GIM severity (mild, moderate, severe) did not differ significantly between these age groups (P = 0.584), with each severity level accounting for approximately one-third of cases in both groups. Similarly, while females slightly outnumbered males, no significant difference in the proportion of GIM severity levels was observed between sexes (P = 0.465), again with each severity level constituting roughly one-third of cases in both males and females. These findings indicate that GIM severity was not significantly associated with age or sex (Table 2).

Table 2 Distribution of GIM severity grades across different age groups and sexes.

Group	Mild (%)	Moderate (%)	Severe (%)	rs/H	P	
Age	<65 years	31 (38.30)	22 (27.20)	28 (34.60)	0.041	0.584	
≥65 years	29 (29.30)	38 (38.40)	32 (32.30)	
Sex	Male	27 (32.14)	26 (30.95)	31 (36.90)	0.533	0.465	
Female	33 (34.38)	34 (35.42)	29 (30.21)	
Notes.

Statistical analyses: Spearman’s rank correlation coefficient (rs) for age group vs. GIM severity; Kruskal–Wallis test (H statistic) for sex vs. GIM severity. P > 0.05 indicates no significant association.

Correlation between GIM subtypes and age/sex

Analysis of GIM subtypes across age groups revealed that the proportion of Type III GIM was slightly higher in the ≥65 years group (28.30%) compared to the <65 years group (22.20%), while the proportion of Type I GIM was slightly lower (18.20% vs. 23.50%, respectively). However, this difference did not reach statistical significance (P = 0.266). Sex distribution analysis showed similar subtype compositions between males and females: Type II predominated (>50%), followed by Type III (∼25%), and Type I (∼20%), with no significant sex difference (P = 0.636) (Table 3).

Table 3 Distribution of GIM subtypes across different age groups and sexes.

Group	Type I (%)	Type II (%)	Type III (%)	rs/H	P	
Age	<65 years	19 (23.50)	44 (54.30)	18 (22.20)	0.083	0.266	
≥65 years	18 (18.20)	53 (53.50)	28 (28.30)	
Sex	Male	19 (22.62)	44 (52.38)	21 (25.00)	0.225	0.636	
Female	18 (18.75)	53 (55.21)	25 (26.04)	
Notes.

Statistical analyses: Spearman’s rank correlation coefficient (rs) for age group vs. GIM subtypes; Kruskal–Wallis test (H statistic) for sex vs. GIM subtypes. P > 0.05 indicates no significant association.

Association between GIM severity and subtype

A significant association was observed between GIM severity and subtype. In mild and moderate GIM, Type III accounted for only approximately 10% of cases, while Types I and II combined represented approximately 90%. In contrast, Type III predominated in severe GIM, accounting for more than 50% of cases, with Types I and II combined representing less than 50%. Spearman correlation analysis confirmed a significant positive correlation between increasing GIM severity and advancing GIM subtype (rs = 0.376, P < 0.05) (Table 4).

Correlation between c-Myc expression and age/sex

c-Myc expression levels showed no significant association with age or sex. The proportion of low c-Myc expression was slightly higher in the ≥65 years group (49.50%) compared to the <65 years group (45.70%), while high expression was slightly lower (13.10% vs. 18.50%, P = 0.449). Males exhibited a marginally higher proportion of low expression (51.19%) and a slightly lower proportion of high expression (14.29%) compared to females (44.79% and 16.67%, respectively; P = 0.401) (Table 5).

Association between c-Myc expression and GIM severity/subtypes

No significant difference in c-Myc expression distribution was found across GIM severity levels (mild, moderate, severe; P = 0.227), although moderate GIM showed a numerically higher proportion of moderate-to-high expression (66.67%) compared to mild (48.33%) and severe GIM (41.67%). Importantly, a significant negative correlation was observed between c-Myc expression and GIM subtype (rs = −0.148, P = 0.047). As the subtype advanced from Type I to Type III: The proportion of low c-Myc expression significantly increased. The proportion of high c-Myc expression significantly decreased. The proportion of moderate expression remained relatively stable (Table 6).

Table 4 Severity and subtype of GIM.

Association between the severity grade of GIM and its histological subtype.

	Type I (%)	Type II (%)	Type III (%)	rs	P	
Mild	20 (33.33)	34 (56.67)	6 (10.00)	0.376	<0.05	
Moderate	8 (13.33)	44 (73.33)	8 (13.33)	
Severe	9 (15.00)	19 (31.67)	32 (53.33)	
Notes.

Statistical analyses: Spearman’s rank correlation coefficient (rs). P < 0.05 indicates a significant positive correlation.

Table 5 Distribution of c-Myc expression levels across different age groups and sexes.

Group	Low (%)	Moderate (%)	High (%)	rs/H	P	
Age	<65 years	37 (45.70)	29 (35.80)	15 (18.50)	−0.057	0.449	
≥65 years	49 (49.50)	37 (37.40)	13 (13.10)	
Sex	Male	43 (51.19)	29 (34.52)	12 (14.29)	0.705	0.401	
Female	43 (44.79)	37 (38.54)	16 (16.67)	
Notes.

Statistical analyses: Spearman’s rank correlation coefficient (rs) for age group vs. c-Myc expression; Kruskal–Wallis test (H statistic) for sex vs. c-Myc expression. P > 0.05 indicates no significant association.

Table 6 Distribution of c-Myc expression in different GIM severities and subtypes.

Association of c-Myc expression levels with GIM severity grade and histological subtype.

Group	Low (%)	Moderate (%)	High (%)	rs	P	
GIM severity	Mild	31 (51.67)	18 (30.00)	11 (18.33)	−0.091	0.227	
Moderate	20 (33.33)	27 (45.00)	13 (21.67)	
Severe	35 (58.33)	21 (35.00)	4 (6.67)	
GIM subtype	Type I	15 (40.54)	13 (35.14)	9 (24.32)	−0.148	0.047	
Type II	45 (46.39)	36 (37.11)	16 (16.49)	
Type III	26 (56.52)	17 (36.96)	3 (6.52)	
Notes.

Statistical analyses: Spearman’s rank correlation coefficient (rs) for both associations. P < 0.05 indicates statistical significance.

Discussion

The incidence of gastric mucosal GIM increases with age, which was confirmed by our study, suggesting that GIM is an age-related pathological change in the gastric mucosa and that aging may be one of its mechanisms. However, this study revealed that the histological severity of GIM (defined as the extent of glandular involvement within a single biopsy sample) did not show a similar trend of aggravation; instead, our results revealed that the proportions of mild, moderate, and severe GIM remained almost constant across the two broad age groups analyzed (<65 and ≥65 years) (for one-third of cases in each group). This result suggests that while GIM incidence rises with age, the histological severity assessed in a single biopsy at a given timepoint may not necessarily progress uniformly with aging within individuals. It also highlights the potential sampling variability inherent in assessing GIM severity from a single biopsy site, which may not capture the full extent or most severe focus of metaplasia present in the stomach.

In this study, type II GIMs were the most common, followed by type III GIMs. Type I GIMs were the least common. While a trend towards a higher proportion of Type III and lower proportion of Type I was observed in the older group (≥65 years) compared to the younger group (<65 years), this difference was not statistically significant. Caution is warranted in interpreting subtype trends across age groups, particularly in the younger subset, given the cohort’s predominant representation of middle-aged and elderly individuals and the small absolute number of very young patients.

Typically, patients and clinicians believe that the more severe the severity of GIM is, the greater the risk of gastric cancer development. This study provides a preliminary answer regarding to what extent the severity of GIM reflects the risk of gastric carcinogenesis. The results revealed that type III GIMs accounted for only approximately 10% of mild-to-moderate GIMs, whereas type III GIMs accounted for more than 50% of severe GIMs. Therefore, we suggest that patients with a severe pathological diagnosis of GIM should be routinely subjected to mucus staining for the GIM subtype to further identify high-risk patients with type III GIM for enhanced gastroscopy follow-up (Shah et al., 2020). This targeted approach could optimize resource allocation in settings where universal mucus staining is not feasible.

Unfortunately, the mucus staining technique required for the GIM subtype has not yet been widely implemented in China. In addition, only a small percentage of patients with type III GIM develop gastric cancer. Therefore, in addition to mucus staining, searching for a tool or marker that can accurately predict the risk of GIM carcinogenesis has long been the focus of gastrointestinal clinical and basic research. In recent years, Operative Link on Gastritis/Intestinal Metaplasia Assessment (OLGA and OLGIM) has been shown to predict the risk of gastric carcinogenesis in atrophic gastritis patients (Rugge et al., 2018; Yue, Shan & Bin, 2018). However, this staging is based on the grading of the severity of atrophy and GIM by the New Sydney system, and the severity of GIM assessed in a single biopsy (as done in this study) did not accurately reflect the GIM subtype distribution or cancer risk associated with subtype. In addition, this staging depends on accurate biopsy taken from both the antrum and the corpus of the stomach according to a standardized protocol (e.g., Sydney protocol). However, in current clinical practice, endoscopists usually take samples from the mucosa of the antrum or angular incisure that seem abnormal under white light or NBI, often resulting in the absence of OLGA or OLGIM staging data, limiting its widespread clinical application.

MYC is a proto-oncogene located on human chromosome (Shah et al., 2020), and one of its expression products, c-Myc, is a transcription factor that directly or indirectly regulates the expression of thousands of genes. c-Myc plays an important role in tumor growth and immune evasion, participates in the development of many human tumors, and is considered the “grand orchestrator” of cancer (Dhanasekaran et al., 2022). The c-Myc gene copy number is abnormally increased in the GIM glands (Kumagai et al., 2022), and single-cell transcriptome analysis revealed that high expression of the c-Myc signaling pathway may be associated with the development of intestinal-type gastric cancer (Huang et al., 2023). Therefore, the feasibility of using c-Myc as a potential risk marker of intestinal-type gastric cancer development should be evaluated by investigating the expression of c-Myc in different severities and subtypes of GIM. In our preliminary study, we found that 80% of patients with mild superficial gastritis (serving as a near-normal comparison) had low expression of c-Myc; however, in our group of GIM patients, moderate-to-high expression accounted for more than 50%, suggesting that c-Myc plays an important role in GIM pathogenesis. However, when we analyzed the relationship between the expression of c-Myc and different severities of GIM, we did not find a positive correlation, suggesting that c-Myc expression is independent of GIM severity. Further analysis of the relationship between different subtypes of GIM and the expression of c-Myc revealed that c-Myc expression decreased when the GIM subtype advanced (from type I to type III). This phenomenon was consistent with the finding that type III GIM increased with increasing severity of GIM. This finding was also the opposite of what we expected based on its oncogenic role and the higher cancer risk associated with Type III GIM. It has been shown that intracellular protein synthesis and degradation are finely regulated at different stages of the cell cycle, which determines the final protein expression level (Alber & Suter, 2019). Therefore, we speculate that there may be a mechanism whereby, although the incidence of GIM is accompanied by an increase in the expression of the MYC gene, the degradation of c-Myc accelerates with the advancement of the GIM subtype (from type I to type III), which may constitute an anticancer protective mechanism in GIM cells. Indeed, BRD4 has been found to play an important role in c-Myc degradation via the ubiquitination of the MYC protein through the phosphorylation of the Thr58 residue (Devaiah et al., 2020). This potential downregulation in advanced subtypes warrants further mechanistic investigation, as it appears counterintuitive to the established pro-oncogenic function of c-Myc. Future studies should integrate genomic (e.g., MYC copy number, mutations) and proteomic analyses to reconcile transcriptional upregulation with potential post-translational regulation leading to decreased protein levels in Type III GIM (Cui et al., 2023). The observed negative correlation between subtype and c-Myc expression should be interpreted considering the limitations of our cohort size and single-center design.

Several limitations should be acknowledged. First, this was a retrospective, single-center study with a limited sample size, which may affect the generalizability of the findings and the power to detect subtle associations, particularly within smaller subgroups (e.g., very young patients). Second, the assessment of GIM severity and subtype was based on a single biopsy per patient, which may not reflect the overall burden or most advanced lesion in the stomach, introducing potential sampling bias (Capelle et al., 2010b). Third, as previously noted, data on H. pylori infection status were unavailable for most patients, precluding analysis of its potential confounding effect on GIM characteristics or c-Myc expression (Huang et al., 2023; Rugge et al., 2018). Fourth, the age groups were broad, and the cohort contained few young individuals, limiting robust analysis of age-related trends, especially for rare subtypes. Finally, the correlative nature of this study precludes establishing causal relationships between GIM subtype progression and changes in c-Myc expression. Prospective, multi-center studies with standardized biopsy mapping (e.g., OLGA/OLGIM protocol), comprehensive H. pylori assessment, and larger sample sizes are needed to validate these findings and explore the underlying mechanisms.

Conclusions

In summary, this study revealed correlations between the expression of c-Myc and the GIM severity and subtype and provided new insights, new suggestions, and ideas for the clinical practice and pathogenesis study of gastric mucosal GIM. Key findings include: (1) While GIM incidence increases with age, the histological severity within a single biopsy does not show a corresponding age-dependent increase; (2) Severe GIM is strongly associated with the Type III (incomplete colonic) subtype; (3) c-Myc expression correlates negatively with advancing GIM subtype (Type I to Type III), despite Type III carrying higher cancer risk, suggesting potential post-translational regulation. Clinically, our results support the selective use of mucus staining in patients diagnosed with severe GIM to identify Type III cases for intensified surveillance. The limitations inherent in this retrospective analysis, particularly the lack of H. pylori data and single-biopsy assessment, should be addressed in future research.

Supplemental Information

Supplemental Information 1 Raw Data

The basic information of 180 patients with gastric intestinal metaplasia.

Supplemental Information 2 Codebook for raw_data.xlsx

We thank LetPub for its linguistic assistance during the preparation of this manuscript.

Additional Information and Declarations

Competing Interests

Author Contributions

Human Ethics

Data Availability

The authors declare there are no competing interests.

Qinglu Yang conceived and designed the experiments, performed the experiments, analyzed the data, prepared figures and/or tables, authored or reviewed drafts of the article, and approved the final draft.

Lingzhi Lian conceived and designed the experiments, performed the experiments, analyzed the data, prepared figures and/or tables, authored or reviewed drafts of the article, and approved the final draft.

Jingying Shen performed the experiments, prepared figures and/or tables, and approved the final draft.

Qin Cao analyzed the data, authored or reviewed drafts of the article, and approved the final draft.

Xuewei Wang conceived and designed the experiments, analyzed the data, prepared figures and/or tables, authored or reviewed drafts of the article, and approved the final draft.

Pingping Hu conceived and designed the experiments, analyzed the data, prepared figures and/or tables, authored or reviewed drafts of the article, and approved the final draft.

The following information was supplied relating to ethical approvals (i.e., approving body and any reference numbers):

The Ethics Committee of Putuo District Center Hospital, Shanghai (Putuo Hospital,Shanghai University of Traditional Chinese Medicine) (Ethical Application Ref: PTEC-A-2025-8(S)-1).

The following information was supplied regarding data availability:

The raw measurements are available in the Supplementary File.

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
