# Peer review of "Gastric intestinal metaplasia subtypes and the effects of c-Myc expression on severity"

_PeerJ, doi:10.7717/peerj.20257_

## Round 0.1 · original submission · Major Revisions

Reviewer 1 ·

Basic reporting

MANUSCRIPT TITLE
Subtype of Intestinal Metaplasia and Expression of c-Myc in Different Severity of Intestinal Metaplasia

AUTHORS
Qinglu Yang, Lingzhi Lian , Jingying Shen , Qin Cao , Xuewei Wang , Pingping

This valuable manuscript (accomplishing all ethics requirements) addresses the relationships between c-MYC expression and gastric IM subtypes.
The authors assessed c-MYC expression (IHC) and IM subtypes (HID) in 180 gastric biopsy samples and concluded that “GIM incidence increases with age; however, GIM severity does not (in a single biopsy sample?). The more severe the GIM is, the greater the proportion of type III GIM cases. c-Myc expression did not correlate with 37 GIM severity. Conversely, as the GIM subtype becomes more advanced, c-Myc expression decreases.” In its present form, the sentence seems to include a contradiction. Could the authors clarify?
The conclusions are interesting and include some novelties.
The biopsy sampling may extensively influence the “severity “ (please specify what “severity“ means) of GIM within the same patient, and such variability affects any conclusion on the cancer risk associated with both the IM subtyping and c-MYC expression.
The methods provide no information about the H. pylori status (current or previous) infection in the address cohort of patients. Such a variable could be relevant in conditioning the achieved results and should be specifically addressed.
Increasing age serves as a proxy for atrophy spreading (including IM) and gastric staging (OLGA and OLGIM). The conclusions should address this potential bias and discuss the possible bias due to the Materials selection.
A negative correlation was observed between the IM subtypes and c-Myc expression (P < 0.05). The loss of c-Myc expression (as the authors say “the grand orchestrator" of cancer”) is interpreted as resulting from “degradation of c-Myc … with the advancement of the GIM subtype (from type I to type III),” which is in contrast to the recognized association between TypII IM and advanced precancerous lesions (i.e., dysplasia) and the literature associating colonic-type IM (types II-III) with increased cancer risk.
In conclusion, while interesting, the manuscript could:
a) Reconsider (specify) the profile of patient selection (H pylori status)
b) Reduce the number of tables. I congratulate on the quality of the histology figures.
c) Reconsider the literature. (among others: PMID: 30138135; PMID: 32117120; PMID: 40108688; PMID: 21538123; PMID: 36739859 PMID: 35772926; PMID: 21481917; PMID: 10894575; PMID: 38383142; PMID: 38207154; PMID: 17142647)
d) Expand the discussion after addressing the two above-mentioned issues.

Experimental design

see above

Validity of the findings

see above

Additional comments

see above

Reviewer 2 ·

Basic reporting

Yang and colleagues present a manuscript addressing the possible correlations between a gastric preneoplastic lesion – gastric intestinal metaplasia (GIM) – its histological subtypes and severity levels. For that they analyse a cohort of 180 patients, focusing on the relationship between GIM subtypes, severity, c-Myc expression, and demographic factors. The study provides some insights into GIM pathology. The observation that GIM severity does not significantly increase with age and that it does not accurately reflect the GIM subtype challenges conventional assumptions and may have important clinical implications. Specific comments:

- There is no legend provided for Figure 1 nor are the different panels (A, B, C...) mentioned in the main text;

- There is no legends for the Tables either;

- Authors should provide a more in-depth discussion of the practical implications for the clinic, especially regarding the recommendation for routine mucus staining in severe GIM cases.

Experimental design

- Please provide more details concerning criteria for patient selection, inclusion/exclusion criteria, and the process for assigning GIM severity, namely for establishing the mild (+), moderate (++), and severe (+++) groups.

Validity of the findings

- The correlations with age are not meaningful for this particular cohort given that most patients are over 60 years old (considered elderly). The youth group only contains 3 patients. Hence, it is difficult with this stratification to sustain the claim that the number of GIM cases increases with age. The same holds true for GIM subtype and severity. It is not accurate to convey the idea that Type III is rare in the youth group when there are only 3 patients in this group. Authors should stratify patients in different age categories (e.g., below 65, over 65) and perform a new analysis.

- Besides cohort size, other limitations related with a single-centre design and potential biases introduced should be discussed.

---

## Round 0.2 · accepted · Accept

Thank you for revising your manuscript to address the reviewers' concerns. Reviewer 2 now recommend acceptance and I am satisfied that the comments of Reviewer 1 have been addressed. The manuscript is now ready for publication.

Reviewer 2 ·

Basic reporting

The Authors have revised the manuscript according to Reviewer's suggestions, particularly concerning caution on interpretation due to age-related bias, and also improved discussion.

Experimental design

Nothing more to add.

Validity of the findings

Nothing more to add.